# Constituents, Enantiomeric Content, and ChE Inhibitory Activity of the Essential Oil from *Hypericum laricifolium* Juss. Aerial Parts Collected in Ecuador

**DOI:** 10.3390/plants11212962

**Published:** 2022-11-02

**Authors:** Melissa Salinas, Nicole Bec, James Calva, Christian Larroque, Giovanni Vidari, Chabaco Armijos

**Affiliations:** 1Departamento de Química, Universidad Técnica Particular de Loja (UTPL), Loja 1101608, Ecuador; 2Institute for Regenerative Medicine and Biotherapy (IRMB), Université de Montpellier, National Institute of Health and Medical Research (INSERM), 34295 Montpellier, France; 3Department of Medical Analysis, Faculty of Applied Science, Tishk International University, Erbil 44001, Kurdistan Region, Iraq

**Keywords:** *Hypericum laricifolium*, essential oil, constituents, enantioselective analysis, AChE and BuChE inhibitory effects

## Abstract

The physical properties, chemical composition, enantiomer distribution, and cholinesterase (ChE) inhibitory activity were determined for a steam-distilled essential oil (EO), with a yield of 0.15 ± 0.05 % (*w/w*), from *H. laricifolium* aerial parts, collected in southern Ecuador. The oil qualitative and quantitative analyses were performed by GC-EIMS and GC-FID techniques, using two capillary columns containing a non-polar 5%-phenyl-methylpolysiloxane and a polar polyethylene glycol stationary phase, respectively. The main constituents (>10%) detected on the two columns were, respectively, limonene (24.29, 23.16%), (*E*)-β-ocimene (21.89, 27.15%), and (*Z*)-β-ocimene (12.88, 16.03%). The EO enantioselective analysis was carried out using a column based on 2,3-diethyl-6-*tert*-butyldimethylsilyl-β-cyclodextrin. Two mixtures of chiral monoterpenes were detected containing (1*R*,5*R*)-(+)-α-pinene (*ee* = 83.68%), and (*S*)-(-)-limonene (*ee* = 88.30%) as the major enantiomers. This finding led to some hypotheses about the existence in the plant of two enantioselective biosynthetic pathways. Finally, the EO exhibited selective inhibitory effects in vitro against butyrylcholinesterase (BuChE) (IC_50_ = 36.80 ± 2.40 µg/mL), which were about three times greater than against acetylcholinesterase (IC_50_ = 106.10 ± 20.20). Thus, the EO from Ecuadorian *H. laricifolium* is an interesting candidate for investigating the mechanism of the selective inhibition of BuChE and for discovering novel drugs to manage the progression of Alzheimer’s disease.

## 1. Introduction

The genus *Hypericum L*., with about 500 known species, is the largest of the nine genera belonging to the family Hypericaceae (order Malpighiales) [1,2], commonly known as the St. John’s wort family. These taxa are distributed worldwide, with native and introduced species, mainly in temperate regions and tropical mountains [3]. Moreover, many species are cultivated as ornamental, medicinal, and timber plants. Several *Hypericum* spp. are used throughout the world in folk medicine, as astringent, febrifuge, diuretic, antiphlogistic, analgesic, antispasmodic, antimigraine, antiepileptic, cholagogue, and antidepressant agents, and to treat diarrhea, dyspepsia, parasite, neuralgia, sciatic, and rheumatism [3,4,5,6]. As a typical example, *H. perforatum* has become one of the most extensively studied herbs and is probably the best-known medicinal herb on the market today [7,8,9]. It is sold in many stores in different commercial forms, such as tinctures, extracts, capsules, ointments, and granules. to treat genitourinary, psychiatric, dermatological, and infectious diseases. However, the results of peer-reviewed medical trials to support the effectiveness of these claims are still debated. In fact, thanks to some clinical studies and meta-analysis reviews supporting the efficacy, St. John’s wort may be prescribed in Germany for mild to moderate depression. On the contrary, in the United States, the plant is considered as only a dietary supplement by the FDA and is not regulated by the same standards as a prescription drug. Moreover, in pastures, St John’s wort acts as an invasive weed and in large doses is poisonous to grazing livestock.

Phytochemical studies carried out on different *Hypericum* taxa have revealed the presence of bioactive specialized metabolites belonging to different classes, including phloroglucinols, napthodianthrones, flavonoids, phenylpropanes, essential oils (EOs), benzophenones, and xanthones [9,10]. Most compounds show a wide range of biological properties, such as antimicrobial [11,12,13], anti-inflammatory [14], anticholinesterase [15,16], anti-cancer [17,18], antioxidant [19,20], anticholinesterase [20], hypoglycemic [21], anti-collagenase and anti-elastase [22], immunosuppressive [23,24], and immunomodulatory effects [25]. For example, the naphthodianthrones hypericin and pseudohypericin, the prenylated phloroglucinol hyperforin, and the flavonoid amentofavone are considered the main compounds responsible for the antidepressant activity of *H. perforatum* [7,8,9].

The *Catalogue of the Vascular Plants of Ecuador* documents 22 accepted *Hypericum* spe-cies known to occur in Ecuador. Of the total figure, 3 species are introduced and, of the 19 native species, 8 are recorded as endemic to Ecuador [26,27]. Interestingly, *H. perfora-tum* is not included in the list.

We have focused our interest on *Hypericum laricifolium* Juss. (Synonyms: *Brathys acer-osa* (*Kunth*) Spach, *B. laricifolia* (Juss.) Spach; *Hypericum acerosum* Kunth; *H. laricoides* Gleason, et al., *H. laricifolium* var. *acerosum* (Kunth) Wedd., *H. platypetalum* Turcz., *H. racemulosum* Turcz. [28]). This species grows abundantly in open or more usually sheltered habitats, at an altitude of 2200–4300 m, from western Venezuela along the Cordilleras Central and Oriental of Colombia and Ecuador to central Peru. Morphologically, it is a small shrub or tree (Figure 1a), from 0.3 m to 3 m tall, bushy, or lax and spreading, with yellow flowers measuring between 15 and 25 mm in diameter, in the shape of a star [28]. 

In the traditional medicine practiced by Ecuadorian communities, *H. laricifolium* is known with different Kichwa and Spanish vernacular names, such as *matikillkana*, *hierba de San Juan*, *romerillo*, *corazoncillo*, *tilín*, *puron* or *gabday*. Especially, the *Kichwa* populations living on Ecuadorian Sierra use an infusion or a decoction as a tranquilizer, to treat headaches, colds, and the flu, to relieve skeletal pain, and in postpartum baths [28]. The plant is also used to combat bad luck and as a protector against evil spirits, as well to dye clothes. Moreover, the whole fresh plant is used as a fragrance in Peru [28].

Investigation on the non-volatile constituents of *H. laricifolium* aerial parts collected in Ecuador has led to the isolation of two new esters of long-chain aliphatic alcohols, hentri-acontanyl and nonacosanyl caffeate, in addition to common sterols and phenolic acids, 3-*epi*-betulinic acid, docosanol, shikimic acid, quercetin, and a few quercetin-3-*O*-glycosides. Quercetin inhibited COX-1 by 44 ± 2% at 200 µM concentration and caffeic acid inhibited COX-2 by 32 ± 16% at 100 µM concentration [29]. In a couple of other publications, several xanthones and one phenyl-γ-pyrone were isolated from aerial parts collected in Venezuela [30] and Peru [31]. One triprenylated xanthone exhibited high activ-ity against the growth of human hepatocellular carcinoma Hep3B cells, with an IC_50_ of 12 μM [31].

Regarding the volatile constituents of *H. laricifolium*, the only essential oil (EO) analyzed so far was hydrodistilled from fresh leaves collected in May near Mérida (Venezuela), at 3900 m above sea level [3]. 

Several contents of *Hypericum* EOs show an extensive qualitative and quantitative variability, even within the same species, depending on different factors, including environmental conditions and geographic distribution [5]. Therefore, we were interested in characterizing the EO from *H. laricifolium* aerial parts collected in Ecuador and in comparing it with the EO from Venezuela [3]. 

Moreover, in continuation of our studies on the cholinesterase (ChE) inhibitory effects of EOs isolated from Ecuadorian plants [32], we examined the activity of the EO from *H. laricifolium* as a cholinesterase inhibitor. ChE inhibitors inhibit the activity of the enzymes acetylcholinesterase (AChE) and butyrylcholinesterase (BuChE) which catalyze the degradation/hydrolysis of the neurotransmitter acetylcholine (ACh) [33,34]. As a result, the level of acetylcholine in neuronal synaptic area increases. Recent studies have demonstrated that in the normal brain, AChE activity predominates over the BuChE activity while, with the progression, for example, of Alzheimer’s disease (AD), brain levels of BuChE increase and AChE activity is maintained or decreases [33,34,35,36]. Thus, both enzymes, being recognized as regulators of ACh levels in the development and progression of AD, represent legitimate therapeutic targets for the symptomatic treatment of this neurodegenerative disease. Their ultimate effect is the amelioration of the cholinergic deficit considered to be responsible for the decline in the cognitive, behavioral, and global functioning characteristics of AD [33,34,35,36]. Currently, a few ChE inhibitors, either synthetic or derived from natural products, such as donepezil, rivastigmine, tacrine or galantamine, are used for the treatment of cognitive dysfunction and memory loss associated with AD [37]. However, these drugs may have adverse effects including gastrointestinal disturbances and problems associated with bioavailability, high cost, and short half-life. Among this class of drugs, Adacanumab, an amyloid-beta directed antibody, was recently approved under an accelerated pathway for the treatment of mild cognitive impairment due to early AD [38]. This drug targets aggregated forms (plaque) of amyloid beta (Aβ) found in the brains of people with AD to reduce its buildup, slowing the progression of the neurodegenerative disease. However, pending the results of the ongoing clinical study on the drug efficacy and the evaluation of the side effects of this new and expensive treatment, there is a constant need for new treatments, reinforcing the interest in finding new ChE inhibitors from natural sources or from the derivatization of natural products [37]. In this context, the essential oils have become of great interest, due to their availability, good penetration of the blood–brain barrier, and few side effects or toxicity, as well as their biodegradability [37,39]. Some extracts of *Hypericum* species, for example, those of *H. perforatum*, *H. humifusum*, *H. neurocalycinum*, and *H. malatyanum*, have exhibited high potential for the treatment of AD [20,40,41]; however, the ChE inhibitory effects of volatile fractions isolated from *Hypericum* species have not yet been tested [5]. 

Finally, to complete the characterization of the EO from *H. laricifolium*, we have determined, for the first time, the enantiomeric composition of an EO from a *Hypericum* species. There is, in fact, great interest to discover new sources of enantiomerically enriched compounds as substrates for stereoselective synthesis; moreover, the pattern of enantiomer distribution may suggest possible biosynthetic paths present in plant cells. 

## 2. Results

### 2.1. Physical Properties of EO

Hydrodistillation from *H. laricifolium* aerial parts (fresh leaves, flowers, and stems), gave a pale yellow EO with an average yield of 0.15 ± 0.05 % (*w/w*) over three replicates. The mean density, refractive index, and specific optical rotation were 0.87 ± 0.01 g/mL, 1.47 ± 0.02, and −14.49 ± 0.05, respectively. 

### 2.2. GC-EIMS and GC-FID Analyses

A total of 46 constituents were identified in the EO of *H. laricifolium* by GC-FID and GC-EIMS using a non-polar DB-5ms and a polar HP-INNOWax column, respectively (Table 1). The main constituents were monoterpenoids, corresponding to 70.25% and 75.80% of the whole EO, respectively. To the entire monoterpenoid fraction, hydrocarbons contributed 67.82% and 75.06% (*w/w*), respectively, whereas oxygenated derivatives were 2.43% and 0.74% (*w/w*), respectively. Sesquiterpenoids accounted for 17.27% and 18.21% (*w/w*), respectively, with hydrocarbons being largely predominant. 

The major identified compounds (>10%) were limonene, (*E*)-β-ocimene, and (*Z*)-β-ocimene, with percentages on the total oil of 24.29, 21.89, and 12.88% on a non-polar DB-5ms capillary column, and 23.16, 27.15, and 16.03 %, on a polar HP-INNOWax capillary column, respectively.

All the quantified constituents accounted for the 98.18% and 98.15% of the whole EO, calculated as the sum of the integrated peak areas of identified compounds with respect to the total peak area in gas chromatograms. The qualitative and quantitative composition of the EO from *H. laricifolium* is reported in Table 1, whereas the GC-EIMS chromatograms are shown in Figure 2 and Figure 3.

### 2.3. Enantioselective Analysis

The enantioselective GC analysis of *H. laricifolium* EO indicated the presence of two pairs of chiral monoterpenes, α-pinene (**1/2**) and limonene (**3/4**) (Figure 4). The enantiomeric distribution and the enantiomeric excesses (*ee*) are shown in Table 2. (*R*)-α-pinene and (*S*)-limonene possessed a high enantiomeric excess, 83.68 and 88.30%, respectively.

### 2.4. Cholinesterase Inhibition Test

The IC_50_ values of the inhibitory activity of the EO from *H. laricifolium* against the cholinesterase enzymes AChE and BuChE were 106.10 ± 20.20 and 36.80 ± 2.40 µg/mL, respectively. Donepezil hydrochloride, used as the positive control, showed an IC_50_ of 0.04 ± 0.02 µg/mL against AChE and 3.60 ± 0.20 µg/mL against BuChE.

## 3. Discussion

With an average yield of 0.15 ± 0.05% (*w/w*), the EO from *H. laricifolium* aerial parts collected in Ecuador confirmed that *Hypericum* is an EO-poor genus, with very low yields (<1%), although sometimes the literature reports higher EO yields, as with other genera, up to 3%, as for the EOs from flowers and leaves of *H. perforatum* [25]. 

The chemical profiles of *Hypericum* EOs exhibit extensive qualitative and quantitative variability, including the distribution of the main constituents [5]. In general, the various parameters affecting the content, composition, and yields of *Hypericum* EOs could be related to the effect of variables such as genetic factors, developmental stages, and seasonal variation phenological cycle, types of plant material and specific organs studied, methods of extraction, environmental conditions, and geographic distribution [5]. The main constituents of the EO from *H. laricifolium* aerial parts collected in Ecuador were limonene (about 26.5%), (*E*)-β-ocimene (about 24.5%), (*Z*)-β-ocimene (about 14.5%). In striking contrast, α-pinene (20.2%), the cembranoid diterpene verticiol (13.4%), 3-methylnonane (12.3%), 2-methyl-octane (9.6%), and *n*-nonane (7.6%) occurred in major amounts in the EO from *H. laricifolium* leaves collected in Venezuela [3]. Such differences in the EO composition between the two samples may thus be attributed to different geographic distribution, types of plant material and organs studied, and seasonal variation. Moreover, the two specimens were collected at different altitudes: at about 2500m a.s.l. in Ecuador and at 3900m in Venezuela. However, the EO differences may also indicate the existence of botanical varieties or subspecies. 

The most frequently reported major constituents of *Hypericum* EOs are the monoter-pene hydrocarbons α- and β-pinenes, the sesquiterpene hydrocarbons (*E*)- β-caryophyllene and germacrene D, and the oxygenated sesquiterpenes spathulenol and caryophyllene oxide; however, in some cases, major constituents are *n*-alkanes, such as *n*-undecane and *n*-nonane [5]. The EO of Ecuadorian *H. laricifolium* belongs to the “terpenoid” type; however, α- and β-pinenes, and germacrene D are only minor components (<3%), while the high content (>12%) of β-ocimene stereoisomers and limonene, accounting for more than 65% of the oil, is unprecedented among *Hypericum* EOs [5]. Such characteristic can thus be considered a reliable identification marker of the EO from Ecuadorian *H. laricifolium* aerial parts.

The results of the GC enantioselective analysis deserve a separate comment. In fact, the finding that a-pinene and limonene occurred in the EO as scalemic mixtures, although one enantiomer predominated in each couple (Table 2), indicated that the biochemical machinery of Ecuadorian *H. laricifolium* contained two separate enzyme systems, each capable of elaborating on a single enantiomer [51]. Thus, the formation of the separate enantiomers of linalyl pyrophosphate (LPP), (*R*)-**6** and (*S*)-**7**, from geranyl pyrophosphate (GPP) (**5**), followed by analogous carbocation reactions, would explain the production of the couples of monoterpene enantiomers **1**/**2** and **3**/**4**, as shown in Figure 1. Moreover, the remarkable higher percent content in the EO of **1**+**4** than **2**+**3** (Table 2), which originated from each of the two LPP enantiomers, clearly indicated that *K*^1^ > *K*^2^ and/or *K*^3^ > *K*^4^. On the higher hand, the higher content of **4** compared to **1**, and **2** compared to **3** (Table 2), which are formed from the corresponding α-terpynyl cation, demonstrated that *K*^5^ is greater than *K*^6^/*K*^7^, but *K*^9^/*K*^10^ are greater than *K*^8^ (Figure 1). In conclusion, these findings clearly indicated that each step of the two biosynthetic pathways leading to **1**-**4** from GPP (**5**) was enzymatically controlled. 

Based on the IC_50_ values (see Section 2.4), the EO from *H. laricifolium* showed higher selective inhibitory activity against BuChE than against AChE. In fact, the former activity was about three times higher than the latter one and about ten times lower than that of the reference inhibitor donepezil hydrochloride. 

The relationship between anticholinesterase activity and structures of oil components, as well as their inhibitory effects mechanism(s), are far from having been firmly determined and even divergent results are observed. These discrepancies often depend on the fact that authors used different experimental parameters for testing cholinesterase inhibitory activity. Undoubtedly a notable trend in the predominant bioactive components is that essential oils with the higher potency for inhibiting AChE activity had monoterpene hydrocarbons, such as α-pinene, β-pinene, δ-3-carene, γ-terpinene, and limonene as their predominant components [37]. Due to the low molecular size and lipophilic nature, they can move into and across cell membranes, and have the potential to influence the fluidity and porosity of membrane structures and to cross the blood–brain barrier. However, polar oxygenated monoterpenoids such as 1,8-cineole, borneol, verbenone, carvacrol, and thymol also exhibited high inhibitory activity against AChE from *Electrophorus electricus* (electric eel) [37]. Moreover, several studies have also attributed the AChE inhibition to the predominant occurrence of sesquiterpenes, especially germacrene-D and α- and β-caryophyllene, in extracts, and the synergistic potential of sesquiterpenes [39]. In fact, although the bio-inhibitory activity of an essential oil could often be explained in terms of the individual effects of some main constituents, other reports have shown that the anti-AChE activity of essential oils is strongly dependent on the interaction of different terpenoid contents, which may include synergy between monoterpenoids and antagonistic relationships between monoterpenoids and sesquiterpenes. For example, limonene can act synergistically with other terpenes to promote terpene absorption by facilitating trans-cellular membrane movement [39].

Recognizing the importance of BuChE in the later stages of AD, interest for BuChE inhibitory activity has recently intensified, although the studies are not as numerous as for the AChE inhibition. It should be noted that 55% of amino acid sequences are identical in the two enzymes; however, 6 of 14 aromatic amino acids contained in the AChE active site are replaced with aliphatic ones in the BuChE active site. Consequently, the BuChE active site is larger in volume than AChE, so that the two ChE enzymes have similar, however distinct, catalytic properties [39]. Monoterpenes α-pinene, sabinene, and limonene showed relatively good inhibitory activity against equine serum BuChE [39]. However, isolated *α*-pinene (7.34 mM) and limonene (0.24 mM) were more than three times less active against equine serum BuChE than against AChE from electric eel [39]. Equine serum BuChE inhibition was also reported for limonene type essential oil isolated from *Crithmum maritimum* L., containing 74.2% of limonene, which showed 24.8% inhibition at 45.5 µg/mL [52]. 

From these findings, it is difficult to attribute the selective higher inhibitory activity of *H. laricifolium* EO against BuChE to the main constituents, limonene and *(Z*)- and (*E*)-β-ocimene. Ocimenes have shown substantial potential as antiviral and antifungal compounds, but there is no information regarding specific anticholinesterase effects. On the other hand, there is still insufficient knowledge about the interactions of the different oil components that can provide clear conclusions about synergistic and antagonistic effects. 

## 4. Materials and Methods

### 4.1. Plant Material

Fresh aerial parts (leaves, flowers, and stems) of *H. laricifolium* (Hypericaceae) (Figure 1b) were harvested in October 2020 from wild plants growing in “Ramos Urku”, Loja province, Ecuador (9588111 N, 692532 E coordinates, at an altitude of 2520 m a.s.l.). The plant was identified by the UTPL botanist Jose Miguel Andrade, and a voucher is deposited in the UTPL Herbarium with the accession code HUTPL11743. The plant material was harvested under permission from the Ministry of Environment of Ecuador (MAE) MAE-DBN-2016-048. Fresh harvested aerial parts were placed in jute bags and brought in about 40 min to the UTPL Chemistry Laboratory, where they were immediately steam-distilled.

### 4.2. Essential Oil Isolation

Fresh aerial parts (1.5 Kg each time) of *H. laricifolium* were subjected to steam-distillation for 3 h using a Clevenger-type apparatus. The process was repeated three times. The essential oils were dried over anhydrous sodium sulfate (Na_2_SO_4_) and stored at −4 °C until analysis. 

### 4.3. Physical Properties of the EO

The relative density of the EO was determined using an analytical balance (Mettler AC 100, Mettler Toledo, Columbus, OH, USA), and a pycnometer of 1 mL capacity according to the AFNOR NF T 75-11 method (ISO 279:1998). The specific optical rotation was determined by an automatic polarimeter (MrcP-810, MRC, Holon, Israel) according to the ISO 592-1998 standard method. Finally, the refractive index was determined by a refractometer (model ABBE, BOECO, Hamburg, Germany) according to the AFNOR NF 75 112 procedure (ISO 280:1998). Each physical property was measured in triplicate at 20 °C, and a mean value was calculated. 

### 4.4. EO Composition Analysis and Sample Preparation

The EO component identification was performed using a coupled Gas Chromatography-Electron Impact Mass Spectrometry (GC-EIMS) instrument whereas, for quantification, a gas chromatography with a flame ionization detector (GC-FID) was employed. For the analyses, a sample of the EO from each distillation (10 µL) was diluted with 990 uL of CH_2_Cl_2_ (dichloromethane). 

#### 4.4.1. GC-EIMS Analysis

The GC-EIMS analysis was performed using an Agilent Technologies GC-MS instrument, consisting of a 6890 N gas chromatograph, equipped with an autoinjector model 7683 and coupled to a mass spectrometer detector (MSD), model 5973 INERT (Santa Clara, CA, USA). The MSD was set in SCAN mode (scan range: *m/z* 40–350), with the electron ionization (EI) source set at 70 eV. Two columns of different polarities were installed: a non-polar DB-5ms column (5%-phenyl-methylpolysiloxane, 30 m × 0.25 mm i.d., 0.25 µm film thickness; J & W Scientific, Folsom, CA, USA) and a polar HP-INNOWax column (polyethylene glycol, 30 m × 0.25 mm i.d., 0.25 µm film thickness; J & W Scientific, Folsom, CA, USA) columns. The oven temperature was increased with a gradient ramp of 3 °C/min from 60 °C to 250 °C, which was maintained for 5 min. The ion source temperature was set at 250 °C, and the carrier gas was helium (GC purity grade from Indura, Guayaquil, Ecuador), set at the constant linear velocity (flow rate) of 1 mL/min. 1 µL of each EO solution was injected in triplicate with a split ratio of 40:1. 

The identity of each EO component was assigned by comparing the corresponding GC-EIMS spectrum and the calculated linear retention index (LRI) with the data reported in the literature. LRIs were calculated in accordance with Van Den Dool and Kratz [53], using a homologous mixture of *n*-alkanes (C_9_-C_25_) injected under the same condition as the *H. laricifolium* EO samples. 

#### 4.4.2. GC-FID Analysis

For the quantitative characterization of the *H. laricifolium* by GC-FID, the operating chromatographic conditions and the chromatographic columns were the same as indicated in the previous paragraph. The percentage of each identified EO component was determined by comparing the area of the corresponding GC peak with the total area of peaks in the gas chromatogram, without applying a correction factor.

### 4.5. Enantioselective Analysis

The enantioselective analysis of the EO was carried out by GC-EIMS, using the same method and instrument configuration as the qualitative analysis, except for the oven temperature program that was as follows: 60 °C for 2 min., followed by a gradient of 2 °C/min until 220 °C, which was maintained for 2 min. A capillary column (25 m × 0.250 mm internal diameter × 0.25 µm phase thickness; Mega, Milan–Italy) was used containing a stationary phase based on 2,3-diethyl-6-*tert*-butyldimethylsilyl-β-cyclodextrin as the chiral selector. The LRIs were calculated in accordance with Van Den Dool and Kratz [53], and the enantiomers’ elution order was determined by injecting enantiomerically pure standards under the same chromatographic conditions. The enantiomer distribution was calculated over three replicates from the peak areas in the gas chromatogram, without applying a correction factor.

### 4.6. AChE and BuChE Inhibition Spectrophotometric Assay

AChE enzyme from *Electrophorus electricus* (Sigma Aldrich, San Luis, MO, USA), and BuChE enzyme from equine serum (Sigma Aldrich, San Luis, MO, USA) were used for the experiments. The AChE and BuChE inhibitory effects of the EO were measured in accordance with the procedure designed by Ellman [54]. A typical inhibition assay volume of 200 µL contained a phosphate buffered saline solution (pH = 7.4), DTNB (5,5-dithiobis-(2-nitrobenzoic acid) ion (1.5 mM), which produces a yellow coloration upon reaction with thiocholine, and the EO sample in DMSO (1% *v/v*). The reaction mixture was preincubated for 10 min at 25 °C; subsequently, acetylcholine iodide (1.5 mM) was added to initiate the reaction. The reaction of DTNB was monitored for 30 min at 30 °C by measuring the absorption at 412 nm in a PherastarFS detection system (BMG Labtech). Absorbance values were collected and expressed as inhibitory concentration (IC_50_) values, which were calculated using the online package GNUPLOT (www.ic50.tk, www.gnuplot.info; accessed on 20 January 2022). Measurements were performed in triplicate. Donepezil was used as the reference ChE inhibitor. False positive results, due to high concentration (>100 ug/mL) of amines or aldehydes, were not excluded [43]. 

## 5. Conclusions

The plant *H. laricifolium* has been widely used by various Andean ethnic groups since ancient times, especially by the *Kichwa* populations of Ecuador. The chemical profile of the essential oil isolated from aerial parts collected in Ecuador was characterized by three major constituents: limonene, (*E*)-β-ocimene, and (*Z*)-β-ocimene, and was thus well differentiated from the EO hydrodistilled from leaves of *H. laricifolium* collected in Venezuela and from other *Hypericum* EOs. Moreover, the enantiomer composition and the activities against the cholinesterase enzymes AChE and BuChE were determined for a *Hypericum* EO for the first time. The selective interesting anti-BuChE properties indicated that the oil is a candidate for investigating the mechanism of the selective BuChE inhibition and for discovering novel drugs to manage the progression of Alzheimer’s disease.

## Data Availability

Not applicable.

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
