# Peer review of "Constituents, Enantiomeric Content, and ChE Inhibitory Activity of the Essential Oil from Hypericum laricifolium Juss. Aerial Parts Collected in Ecuador"

_plants, 2022, doi:10.3390/plants11212962_

Round 1
Reviewer 1 Report
I think that the authors have been successful and that the manuscript should be considered for publication in this journal as it also fits within the scope and aims of Plants and would be of interest to a readership interested in interested in applying essential oils from H. laricifolium and their mechanism of the selective BuChE inhibition and for discovering novel drugs to manage the progression of Alzheimer’s disease.
Author Response
Dear Reviewer, thank you very much for your comments on our article

Reviewer 2 Report
- Chemical composition, - I notice that only the aromas are determined, I suggest you to improve with other determinations considering that in the title you specify "chemical composition"
- Plant material - I recommend a more detailed description of the storage method after harvesting, if it was stored, under what conditions? how much time?
-
Author Response
Reply to Reviewer 2: In the last version of the manuscript, the changes suggested are in pale blue color.
Chemical composition, - I notice that only the aromas are determined, I suggest you to improve with other determinations considering that in the title you specify "chemical composition"
Answer: Dear reviewer, thank you for your comment. In the title of the article and the keywords we have replaced “chemical composition” by “constituents”, which are now in pale blue colour. In this manner we indicate that in this study, we have only studied the chemical composition of essential oil.
- Plant material - I recommend a more detailed description of the storage method after harvesting, if it was stored, under what conditions? how much time?
Answer: Dear Reviewer, thank you very much for your comments on our article. In the point 4.1: Plant Material, we have added new information.
With my kindest regards,
Chabaco Armijos, Dr.
Correspondence: cparmijos@utpl.edu.ec; Tel.: +593-07-370-1444
